# CPAP Treatment at Home after Acute Decompensated Heart Failure in Patients with Obstructive Sleep Apnea

**DOI:** 10.3390/jcm13195676

**Published:** 2024-09-24

**Authors:** Petar Kalaydzhiev, Angelina Borizanova, Neli Georgieva, Gergana Voynova, Slavi Yakov, Tsvetan Kocev, Galya Tomova-Lyutakova, Bozhidar Krastev, Natalia Spasova, Radostina Ilieva, Elena Kinova, Assen Goudev

**Affiliations:** 1Department of Emergency Medicine, Medical University—Sofia, 1000 Sofia, Bulgaria; 2Cardiology Department, University Hospital “Tsaritsa Yoanna—ISUL”, 1000 Sofia, Bulgaria

**Keywords:** acute decompensated heart failure, obstructive sleep apnea, continuous positive airway pressure therapy

## Abstract

**Background**: Acute decompensated heart failure (ADHF) is a condition with a high frequency of hospitalizations and mortality, and obstructive sleep apnea (OSA) is a common comorbidity. Continuous positive airway pressure (CPAP) therapy at home can be a good adjunctive non-drug therapy for these patients. **Methods**: We conducted a single-center, prospective cohort study from 150 consecutive patients hospitalized for heart failure exacerbation in the cardiology department. Of these, 57 patients had obstructive sleep apnea. After discharge, CPAP therapy at home was offered. We divided them into two groups and followed them for 1 year. All patients received optimal medical treatment. At the end of the period, patients underwent a follow-up physical examination, a follow-up echocardiography, and a follow-up evaluation of the Epworth Sleepiness Scale (ESS). **Results**: From 81 patients with sleep apnea, 72.8% (n = 59) had obstructive sleep apnea (OSA) and 27.2% (n = 22) had central sleep apnea (CSA). There was a statistically significant difference in body mass index (BMI), ESS, systolic blood pressure (SBP), diastolic blood pressure (DBP), and left ventricular ejection fraction (LVEF%) in the group with CPAP therapy compared to the no-CPAP group. The CPAP group had a median survival of 11.7 months vs. 10.1 months in the no-CPAP group (log-rank (Mantel–Cox) *p* = 0.044). **Conclusions**: This study suggests that obstructive sleep apnea is a common comorbidity in patients with acute decompensated heart failure. The addition of CPAP therapy in these patients improves the symptoms and the prognosis.

## 1. Introduction

ADHF is a major contributor to hospitalization and mortality in both the United States and Europe, underscoring its significant burden on healthcare systems [1,2]. Sleep-disordered breathing (SDB) is highly prevalent among patients with ADHF but is often underdiagnosed despite its profound impact on patient outcomes [3]. Obstructive sleep apnea and central sleep apnea are the two primary forms of SDB observed in these patients, each with distinct pathophysiological mechanisms and implications for heart failure progression [4]. OSA is more frequently detected in patients with ADHF and is characterized by repetitive episodes of upper airway obstruction during sleep, leading to intermittent hypoxia and sympathetic activation [5]. Reduced LVEF% in patients with OSA has been linked to increased left ventricular filling pressures and higher pulmonary artery pressures, which exacerbate heart failure symptoms and contribute to a worse prognosis [6]. Additionally, elevated E/e’ ratios in OSA patients suggest impaired diastolic function [7]. CSA is more commonly associated with significantly reduced LVEF%, reflecting a more severe form of heart failure compared to OSA [5]. Patients with CSA often exhibit higher E/e’ ratios, indicating more pronounced diastolic dysfunction, and they tend to have elevated NT-proBNP levels due to the combined effects of ventricular wall stress and frequent episodes of hyperventilation and apnea [8,9]. The differences in pathophysiology between OSA and CSA highlight the need for tailored therapeutic approaches in patients with ADHF. CPAP therapy has been shown to improve outcomes in patients with chronic heart failure (CHF) and OSA by reducing sympathetic activity, lowering blood pressure, and improving left ventricular function [10,11,12,13]; however, the efficacy of CPAP in patients with CSA remains controversial as it may not address the underlying central respiratory instability [14]. Recent studies have begun to explore the role of CPAP therapy in patients with ADHF, particularly those with newly diagnosed OSA, showing promising results in terms of reducing hospitalization and mortality [15]. However, there is a significant gap in the literature regarding the long-term use of CPAP therapy at home for these patients. Addressing this gap is critical, as long-term management strategies that consider the distinct pathophysiological mechanisms of OSA and CSA could lead to better outcomes in patients with ADHF and concomitant sleep apnea.

## 2. Aim

To determine the frequency of obstructive sleep apnea in patients with acute decompensated heart failure, our study aimed to incorporate home CPAP therapy alongside standard medical treatment for heart failure. We sought to evaluate the impact of this combined approach on patient outcomes over a one-year period, focusing on clinical assessments and mortality rates. By tracking these patients throughout the year, we aimed to gain a comprehensive understanding of how adding CPAP therapy to conventional treatments affects the overall management of ADHF and improves patient survival and quality of life.

## 3. Materials and Methods

### 3.1. Study Design

We undertook a single-center, prospective cohort study involving a total of 150 consecutive patients who were admitted to the cardiology department of University Hospital “Tsaritsa Yoanna—ISUL” in Sofia for the management of acute decompensated heart failure. This study spanned a period of three years, from January 2017 to December 2019. Out of these 150 patients, 81 individuals met the specific inclusion and exclusion criteria that are meticulously defined and outlined in Table 1. These criteria were established to ensure the selection of a well-characterized patient cohort, thereby facilitating the robustness and generalizability of our findings. The inclusion criteria encompassed detailed clinical, demographic, and laboratory parameters, while the exclusion criteria were carefully delineated to exclude patients with conditions or factors that could confound the study results.

### 3.2. Study Participants

During the hospital stay, after obtaining informed consent—approved by the Institutional Ethics Committee of the Medical University of Sofia (protocol code 2100 from 27 April 2016)—patients with an Epworth Sleepiness Scale (ESS) score above 8 underwent screening for sleep apnea using the ApneaLink™ system (ResMed, San Diego, CA, USA). The ApneaLink™ is a portable device used to detect both obstructive and central sleep apnea by monitoring physiological parameters during sleep including airflow, chest movements, heart rate, and blood oxygen levels. This system has been shown to be an effective alternative to full polysomnography (PSG) for initial sleep apnea assessment, particularly in home settings. Several studies have demonstrated a strong correlation between the AHI values derived from ApneaLink™ and those obtained from PSG. Although PSG remains the gold standard for diagnosing sleep apnea, devices like ApneaLink™ provide a more accessible and convenient method for preliminary screening, making it a valuable tool in clinical practice [16]. Automatic CPAP devices with full-face masks were used for the therapy. These devices allowed continuous online monitoring. Usage data were uploaded nightly to a shared platform, enabling tracking and encouraging adherence to therapy. Our goal was for patients to use the devices for at least 4 h per night. Monthly reports were sent to the patients, and we maintained regular contact with them.

The AHI, representing the number of apneas and hypopneas per hour of sleep, was calculated for each patient. Data collected by the ApneaLink™ device were analyzed using its dedicated reporting software to determine whether patients had OSA or CSA.

Patients were then divided into two groups based on their diagnosis: those with OSA (n = 59) and those with CSA (n = 22). Furthermore, within the OSA group, patients were offered the option of CPAP therapy at home following hospital discharge. This resulted in the creation of two additional subgroups: patients who chose to continue CPAP therapy (n = 17) and those who did not (n = 42).

All participants in the study were followed over the course of 12 months, with the primary endpoints being all-cause mortality and heart-failure-related hospitalizations.

### 3.3. Statistical Analysis

Continuous variables were expressed as means with standard deviations to describe central tendencies and variability in the data. For categorical variables, frequency analysis was conducted to summarize and understand the distribution across the study population. The chi-squared (χ^2^) test was employed for the comparison of qualitative variables, enabling the identification of potential associations between categorical data points. In cases where sample sizes were small, adjustments such as continuity corrections were applied to enhance the reliability of the results. 

The Kolmogorov–Smirnov test was utilized to assess whether the distribution of the data followed a normal pattern. Based on the normality results, appropriate statistical tests were chosen. When comparing two independent groups, Student’s *t*-test was applied for normally distributed data, while the Mann–Whitney U test was used for non-normally distributed data. These tests allowed for determining whether statistically significant differences existed between the two groups. When comparing more than two groups, analysis of variance (ANOVA) with Bonferroni’s correction was used to account for multiple comparisons and reduce the risk of false positives. In cases where the assumption of normality was not met, non-parametric alternatives, such as the Kruskal–Wallis test, were employed to compare medians across groups. Pearson’s correlation coefficient was used for normally distributed quantitative data, while Spearman’s rank correlation was applied for non-normally distributed data to assess linear relationships between variables. 

Kaplan–Meier survival curves were generated to estimate survival probabilities and re-hospitalization risks, with the log-rank test applied for group comparisons. Statistical significance was considered at *p* < 0.05. All analyses were conducted using SPSS version 24, with graphical representations created using Microsoft Excel (Version 16.0) and IBM SPSS Statistics for Windows (Version 24.0) to ensure clarity and accuracy.

## 4. Results

### 4.1. Patient Demographics and Clinical Characteristics 

In this study, we classified the 81 patients into two groups based on the type of sleep apnea they had: obstructive sleep apnea or central sleep apnea. Among these patients, 59 (72.8%) were diagnosed with OSA, while 22 (27.2%) had CSA. The average age of patients with OSA was 67 years, compared to an average age of 69 years in the CSA group.

Regarding gender distribution, our analysis showed no significant differences between the two groups. The proportion of male and female patients was similar in both the OSA and CSA cohorts. Detailed clinical characteristics and comorbidities for each group are outlined in Table 2.

### 4.2. OSA Patients

For the patients with obstructive sleep apnea, we divided them into two groups based on their post-discharge treatment: those who received CPAP therapy and those who did not. Out of the 59 OSA patients, 17 were prescribed CPAP, while 42 did not receive this therapy.

We followed up with these patients over a 12-month period after their discharge. At the end of this period, each patient underwent a follow-up physical examination, an echocardiogram, and a reassessment using the Epworth Sleepiness Scale. Through telemedicine, we monitored the data from the automatic CPAP devices, which allowed us to remotely adjust the pressure settings for optimal comfort. This approach ensured that every patient was able to use the device for at least 4 h each night, meeting our adherence goal for all participants.

In the group that did not receive CPAP, 29 patients completed the follow-up, whereas 13 were lost to follow-up due to mortality. In the CPAP group, 16 patients were followed up, with 1 patient dropping out due to a fatality.

This follow-up process provided insights into the effectiveness of CPAP therapy and its impact on patient outcomes over time.

#### 4.2.1. BMI Follow-Up in the CPAP and Non-CPAP Groups

In our comparative analysis of body mass index changes between the two groups of patients—those who were treated with CPAP therapy during sleep and those who did not receive CPAP therapy—we observed some significant differences. For the patients who used CPAP therapy, we noted a substantial reduction in BMI from the beginning to the end of the follow-up period. Initially, the average BMI of these patients was 36.47 ± 7.3. By the conclusion of the follow-up, this average BMI had decreased to 33.3 ± 5.7, with a *p*-value of less than 0.001, indicating a statistically significant change.

On the other hand, for patients who did not receive CPAP therapy, the changes in BMI were not statistically significant. Their average BMI at the start of the follow-up was 36.35 ± 7.34, which changed slightly to 35.74 ± 7.46 by the end of the follow-up period. The *p*-value for this comparison was 0.356, suggesting that the difference observed was not significant. The results are presented in Figure 1. 

#### 4.2.2. Comparison Regarding Daytime Sleepiness

The assessment of daytime sleepiness was performed using the Epworth Sleepiness Scale, and the results revealed a notable difference between the two groups. For patients who received CPAP therapy, there was a significant improvement in ESS scores over the study period. Initially, the average ESS score for the CPAP group was 12.4 ± 2.4. By the end of the follow-up, this score had significantly decreased to 6.3 ± 1.9, with a *p*-value of less than 0.001, indicating a highly significant improvement in daytime sleepiness.

In contrast, the group of patients who did not receive CPAP therapy did not exhibit a similar improvement. Their average ESS score was 12.0 ± 2.6 at the beginning of the study and changed minimally to 11.59 ± 2.8 by the end. The *p*-value for this comparison was 0.19, suggesting that the change in daytime sleepiness was not statistically significant for this group.

These results underscore the effectiveness of CPAP therapy in alleviating daytime sleepiness, as reflected by the significant reduction in ESS scores among patients receiving the treatment, compared to those who did not. Figure 2 displays these changes. 

#### 4.2.3. Comparison in Terms of Systolic and Diastolic Blood Pressure

We conducted a comparative analysis of systolic blood pressure (SBP) and diastolic blood pressure (DBP) by evaluating the mean values at both the beginning and the end of the observation period.

For systolic blood pressure, we observed a statistically significant reduction in the group that received CPAP therapy. Initially, the mean SBP in this group was 136.2 ± 6.2 mmHg. By the end of the follow-up period, it had decreased to 123.1 ± 5.9 mmHg, with a *p*-value of less than 0.001, indicating a highly significant decrease.

In contrast, the group that did not receive CPAP therapy showed no significant change in systolic blood pressure. Their mean SBP was 131.2 ± 9.8 mmHg at the start of the study and remained virtually unchanged at 131.3 ± 7.7 mmHg at the end, with a *p*-value of 0.965, suggesting that the difference was not statistically significant.

These findings highlight the positive impact of CPAP therapy on reducing systolic blood pressure, whereas no such effect was observed in patients who did not receive CPAP treatment. The results are depicted in Figure 3.

We also assessed the changes in diastolic blood pressure (DBP) over the study period for both patient groups. The results showed a significant decrease in DBP for the cohort receiving continuous positive airway pressure (CPAP) therapy. Initially, the mean DBP in the CPAP group was 85.9 ± 6.7 mmHg. By the end of the follow-up period, this value had dropped to 77.3 ± 5.3 mmHg, with a *p*-value of less than 0.001, indicating a statistically significant reduction.

In contrast, the group that did not receive CPAP therapy did not experience a significant change in DBP. The mean DBP in this group was 82.1 ± 7.4 mmHg at the start of the study and changed minimally to 81.7 ± 5.7 mmHg by the end, with a *p*-value of 0.644. This result suggests that the non-CPAP group did not show a statistically significant difference in DBP over the observation period. These findings are illustrated in Figure 4. 

#### 4.2.4. Change in LVEF% in Patients with and without CPAP Therapy

We observed an improvement in ejection fraction among patients who received CPAP therapy. In the CPAP group, the average LVEF% increased from 47.5 ± 8.7% at the beginning of the study to 50.07 ± 7.5% by the end of the follow-up period. This change was statistically significant, with a *p*-value of 0.005, indicating a meaningful improvement in cardiac function.

Conversely, the group that did not receive CPAP therapy did not show a significant change in LVEF%. Their average LVEF% was 52.5 ± 6.9% at the start and slightly decreased to 52.2 ± 7.1% after 12 months. The *p*-value for this comparison was 0.389, suggesting that the difference was not statistically significant.

These results underscore the positive impact of CPAP therapy on improving the ejection fraction, highlighting its potential benefits for enhancing cardiac function in patients with obstructive sleep apnea. Figure 5. 

#### 4.2.5. Survival Analysis

Our analysis of survival outcomes revealed that patients with OSA who used CPAP therapy had a significantly longer survival time compared to those who did not receive CPAP. On average, patients in the CPAP group lived for 11.7 months, while those in the non-CPAP group had an average survival time of 10.1 months. This difference was statistically significant, as shown by the Kaplan–Meier survival curves and confirmed by the log-rank test, which had a *p*-value of 0.044.

The Kaplan–Meier curves clearly demonstrate that the CPAP group had a better survival rate over time compared to the non-CPAP group. The log-rank test supports this observation, indicating that the improved survival associated with CPAP therapy is statistically significant. These findings underscore the significant advantage of CPAP therapy, not just in alleviating the symptoms of obstructive sleep apnea, but also in improving overall patient survival. Figure 6 displays the curves.

## 5. Discussion

The distribution of predominant sleep apnea types among patients with acute decompensated heart failure exhibits considerable variability, reflecting the inherent complexity of this patient population. In our study, we identified that approximately 73% of these patients have OSA, while 27% suffer from CSA. These findings are in line with the broader literature, which reports a prevalence of OSA ranging from 21% to 80% and CSA from 15% to 45% in similar patient cohorts [4,17,18]. This variability in prevalence rates can largely be attributed to differences in the types and severities of heart failure among the studied populations [19]. Our study further highlighted significant differences in key clinical parameters such as LVEF%, the E/e’ ratio, and levels of NT-proBNP. Specifically, patients with reduced systolic function—evidenced by a lower LVEF% and elevated NT-proBNP levels—were found to have lower left ventricular filling pressures and higher pulmonary pressures, making them more susceptible to hyperventilation and Cheyne–Stokes respiration during sleep [18,19]. This aligns with the observed tendency for patients with more severe heart failure to experience these specific types of sleep disturbances. Additionally, our research identified obesity and arterial hypertension as significant risk factors for developing OSA. This observation supports the established pathophysiological mechanisms linking OSA to cardiovascular risk, including neurohormonal dysregulation, endothelial dysfunction, and systemic inflammation [20]. These relationships remained evident in our ADHF patient cohort, as further detailed in Table 2. Understanding these associations is crucial for tailoring treatment strategies and managing the complexities of sleep apnea in patients with acute decompensated heart failure.

The use of home continuous positive airway pressure therapy has shown promising outcomes in patients with heart failure, particularly in improving body mass index, reducing daytime sleepiness, and enhancing overall quality of life [21]; however, its benefits have not been extensively studied in patients with ADHF, despite its established efficacy in CHF populations [22,23,24]. Our study highlights the potential benefits of CPAP therapy in this group. However, the small sample size of the CPAP group (only 17 participants) is a significant limitation. The limited number of participants reduces the statistical power, making the results less robust and harder to generalize to a broader population. Larger future studies are needed to confirm these findings, provide greater representativeness, and yield more reliable data regarding the efficacy of CPAP therapy. Recruiting ADHF patients for clinical trials remains challenging, and randomized clinical trials (RCTs) on this subject are limited. The 2016 SAVE trial by McEvoy RD et al. is one of the few large-scale RCTs available, which reported that CPAP therapy did not significantly reduce cardiovascular events, including heart failure, although it did improve daytime sleepiness, reduce weight, and enhance quality of life [25]. Furthermore, CPAP therapy was associated with significant reductions in both systolic and diastolic blood pressure. These findings are corroborated by a 2015 RCT conducted by Gottlieb DJ et al., which included 288 patients and observed similar beneficial effects over a 12-week follow-up period [26].

The necessity of integrating CPAP therapy into the standard treatment regimen for heart failure patients remains a subject of ongoing debate. While small-scale studies have demonstrated improvements in left ventricular systolic and diastolic function with CPAP therapy [27,28,29], larger RCTs have failed to show significant changes in these parameters [25]. These discrepancies can be attributed to several factors, including variations in follow-up periods (ranging from 6 months to 2 years) and differences in patient selection criteria. Some studies exclusively included patients with reduced systolic function, while others, including our study, did not impose such criteria. These methodological differences likely contribute to the observed inconsistencies across studies. Notably, our study documented a significant improvement in LVEF with CPAP therapy, which is consistent with a 2020 RCT by Khayat et al. involving 150 ADHF patients; their study reported an increase in ejection fraction associated with CPAP therapy administered during hospitalization [5]. These findings support the hypothesis that CPAP therapy is an effective and safe non-pharmacological treatment for patients with OSA and heart failure. Similar improvements in LVEF have also been reported by Naito R et al. in small patient cohorts [30]. Additionally, a randomized study by Kim D, involving 52 patients, compared sham CPAP with real CPAP therapy over a 3-month period, finding greater improvements in LVEF% in the real CPAP group [31]. A comprehensive 2018 meta-analysis further supports the benefits of CPAP therapy, particularly its role in preventing major cardiovascular events [32]. This meta-analysis synthesized data from seven RCTs, encompassing a total of 4268 patients, thus providing robust evidence in favor of CPAP therapy.

The use of non-invasive home ventilation, such as continuous positive airway pressure therapy, as an adjunctive non-pharmacological treatment for heart failure is an area of considerable interest. Despite its potential benefits, clinical trials have often reported mixed outcomes, particularly regarding overall and cardiovascular mortality [24,33,34]. In contrast, our study demonstrated a significantly higher survival rate among patients receiving CPAP therapy. Telemedicine, monthly contact, and strict monitoring of data from the automatic CPAP therapy may have contributed to the lower mortality rate in the non-invasive ventilation group. This finding aligns with a study by Khayat R et al., which observed the lowest mortality rates in a subgroup of patients who received CPAP therapy post-discharge, closely resembling our patient cohort [5]. 

However, it is important to recognize the limitations of our study. Notably, the study was conducted prior to the widespread adoption of newer heart failure therapies, such as sacubitril/valsartan and SGLT2 inhibitors. These medications have since become standard components of heart failure treatment and may have influenced patient outcomes. The absence of these therapies during our study and follow-up periods represents a significant confounding factor.

Recent RCTs are investigating the effects of these newer therapies on patients with sleep apnea. Preliminary results suggest that sacubitril/valsartan and SGLT2 inhibitors may provide additional benefits, particularly when used in conjunction with CPAP therapy [31,35,36]. These findings could potentially enhance our understanding of how to optimize treatment strategies for heart failure patients with sleep apnea. Future research incorporating these advanced therapies will be crucial for further elucidating their impact and refining treatment approaches for this complex patient population.

## 6. Future Directions

To build on the insights from our study, future research should involve larger, multi-center trials to ensure that the findings are applicable to a broader population. Expanding the research to include more diverse patient groups will help us understand the effects of CPAP therapy on OSA and its related conditions more comprehensively.

It is also important to explore the mechanisms that link OSA to heart failure. Understanding how CPAP therapy influences cardiovascular outcomes could provide valuable insights and lead to more effective treatments.

In addition, future studies should focus on monitoring CPAP adherence more closely. By examining how well patients stick to their CPAP therapy and its impact over time, researchers can gain a better understanding of its long-term effectiveness and how it affects patient outcomes.

Overall, these steps will help refine our treatment approaches, improve patient care, and deepen our understanding of how OSA and cardiovascular health are connected.

## 7. Conclusions

In conclusion, OSA often occurs alongside acute decompensated heart failure, and integrating home CPAP therapy with standard pharmacological treatments has proven to be highly beneficial. Specifically, CPAP therapy has been associated with a range of positive outcomes, including reductions in BMI and improvements in daytime sleepiness. Additionally, patients receiving CPAP therapy experienced significant decreases in both systolic and diastolic blood pressure, enhancements in left ventricular systolic function, and increased survival rates.

These benefits suggest that CPAP therapy could play a crucial role in managing ADHF by addressing several critical aspects of the condition. However, to confirm these findings and fully establish the role of CPAP therapy in the treatment of ADHF, it is essential to conduct further research. Well-designed randomized clinical trials will be necessary to validate these results and provide clearer guidelines on the optimal use of CPAP therapy in this context. 

## Figures and Tables

**Figure 1 jcm-13-05676-f001:**
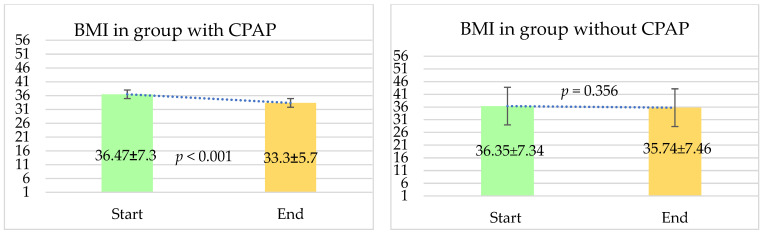
BMI at the start and at the end of the study in patients with and without CPAP. The follow-up period was 12 months. BMI—body mass index; CPAP—continuous positive airway pressure. Results are expressed as mean ± SD.

**Figure 2 jcm-13-05676-f002:**
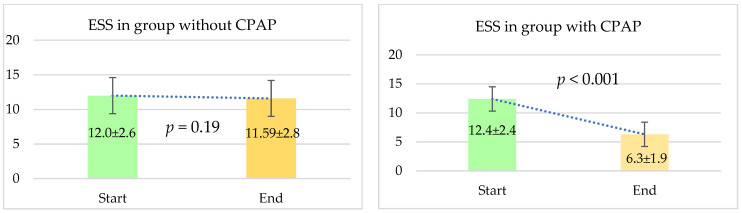
Change in ESS from baseline to end of follow-up in the CPAP and non-CPAP groups. The follow-up period was 12 months. ESS—Epworth Sleepiness Scale. Results are expressed as mean ± SD.

**Figure 3 jcm-13-05676-f003:**
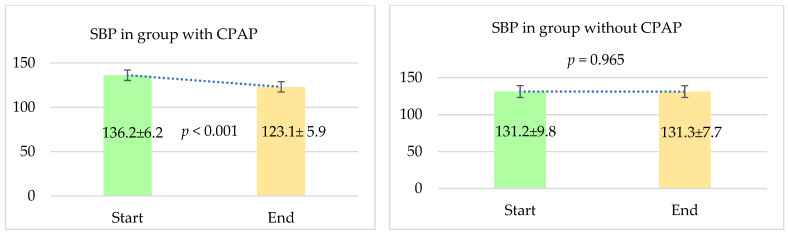
Comparison in terms of systolic blood pressure (SBP) between two groups. The follow-up period was 12 months. Results are expressed as mean ± SD.

**Figure 4 jcm-13-05676-f004:**
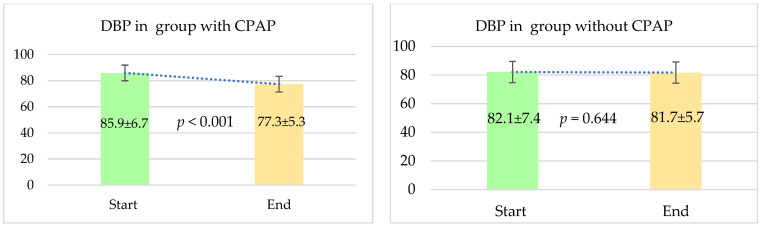
Comparison in terms of diastolic blood pressure (DBP) between two groups. The follow-up period was 12 months. Results are expressed as mean ± SD.

**Figure 5 jcm-13-05676-f005:**
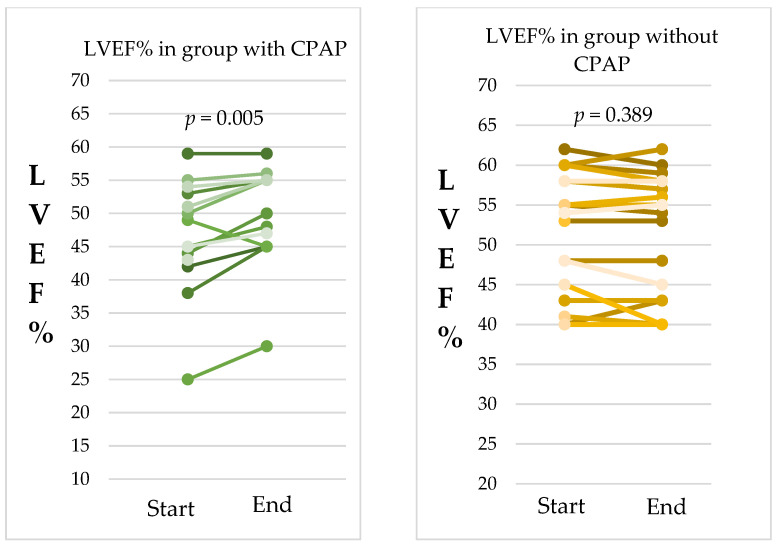
Change in left ventricular ejection fraction (LVEF%) in patients with and without CPAP therapy. The follow-up period was 12 months.

**Figure 6 jcm-13-05676-f006:**
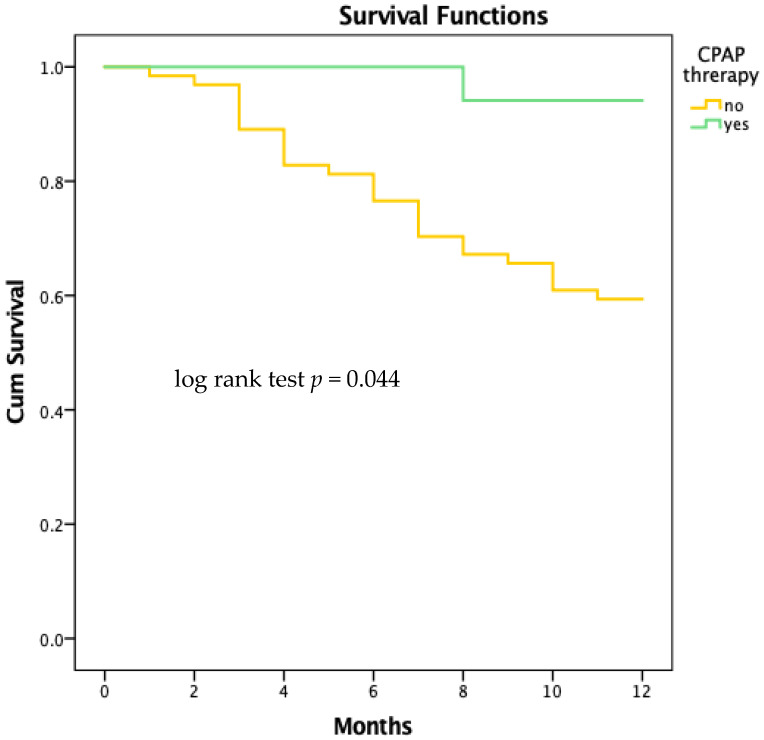
Kaplan–Meier survival curves for patients with CPAP therapy and without CPAP therapy. CPAP—continuous positive airway pressure.

**Table 1 jcm-13-05676-t001:** Inclusion and exclusion criteria.

Inclusion Criteria	Exclusion Criteria
Clinical manifestations of HF New York Heart Association (NYHA) class II/III Daytime sleepiness questionnaire of Epworth Sleepiness Scale (ESS) > 8 N-Terminal Fragment of the Prohormone Brain-Type Natriuretic Peptide—(NT-proBNP) > 900 pg/mL Apnea–hypopnea index (AHI) > 5 Signed informed consent	New York Heart Association (NYHA) class IV or terminal HF Acute respiratory failure Acute coronary syndrome End-stage kidney insufficiency Terminal liver insufficiency Chronic respiratory diseases (COPD) Unsigned informed consent

**Table 2 jcm-13-05676-t002:** Patients’ clinical characteristics.

Clinical Characteristic	OSA n = 59	CSA n = 22	*p*–Value
Age, years	67.08 ± 8.840	69.05 ± 9.62	0.382
Gender, male %	57.6	54.5	0.499
Arterial hypertension, %	74.6	68.1	0.565
Ischemic heart disease, %	47.5	86.3	**0.002**
Left ventricular hypertrophy, %	79.7	50	**0.008**
Diabetes mellitus, %	74.5	40.9	**0.005**
Atrial fibrillation, %	49.1	81.8	**0.008**
SBP, mm Hg	132.7 ± 89.07	111.04 ± 811.1	**<0.001**
DBP, mm Hg	83.3 ± 87.08	71.5 ± 87.9	**<0.001**
Mean heart rate, beats/min	73.1 ± 88.2	86.9 ± 88.9	**<0.001**
BMI	36.8 ± 87.07	31.9 ± 83.5	**0.002**
NT-proBNP, pg/mL	1623 ± 8897	3500 ± 81,453	**<0.001**
LVEF%	49.6 ± 8.04	42.5 ± 9.4	**<0.001**
E/e’m	16.5 ± 3.8	18.6 ± 2.9	**0.016**
AHI	42.3 ± 22.3	34.8 ± 9.8	0.129

Abbreviations: OSA, obstructive sleep apnea; CSA, central sleep apnea; SBP, systolic blood pressure; DBP, diastolic blood pressure; BMI, body mass index; NT-proBNP, N-terminal pro–B-type natriuretic peptide; LVEF, left ventricular ejection fraction; E/e’m, peak E-wave to peak e’ wave ratio; AHI, apnea–hypopnea index. Results are expressed as mean ± SD.

## Data Availability

The data underlying the findings of this study are subject to data protection regulations and privacy laws. As such, they are not publicly available. However, the data can be accessed upon reasonable request, contingent upon compliance with all relevant confidentiality and privacy requirements.

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
