# Peer review of "CPAP Treatment at Home after Acute Decompensated Heart Failure in Patients with Obstructive Sleep Apnea"

_jcm, 2024, doi:10.3390/jcm13195676_

Round 1
Reviewer 1 Report
Comments and Suggestions for Authors
Improve the layout (spaces, double spaces etc)
Line 13 Patients with obstructive sleep apnea (OSA) are common comorbidities. Improve English
Line 15 for these. For them
Line 21 From 81 with sleep apnea 72.8% (n=59) has obstructive sleep apnea 21 (OSA) and 27.2% (n=22) has central sleep apnea (CSA). Incomprehensible
Line 37 To a greater extent, obstructive sleep apnea is detected Incomprehensible
Discuss all the alternatives in treating OSA.
Explain how the patients were recruited.
Comments on the Quality of English LanguageExtensive editing of English language required.
Author Response
Thank you very much for your valuable recommendations and insightful feedback on our manuscript. We greatly appreciate your recognition of the strengths of our study, as well as the specific areas where we can enhance the presentation of our findings.
We have carefully reviewed your comments and made the necessary revisions to improve the clarity, structure, and depth of our analysis. Additionally, we have addressed the potential limitations of our study and included suggestions for future research, as you advised.
Once again, thank you for your constructive feedback. We believe the revisions have significantly strengthened our manuscript, and we look forward to resubmitting it for the next stage of the review process.
Reviewer 2 Report
Comments and Suggestions for Authors
Article
CPAP treatment at home after acute decompensated heart failure in patients with obstructive sleep apnea.
Authors present an over-all well written study on the benefit of home CPAP treatment for patients with obstructive sleep apnea and heart failure. Amongst the significant clinical differences found between CPAP/non-CPAP patients was a marked increase in survival for those using CPAP within a 12 month period. This study will be of particular interest to those in the field and is recommended for publication after a few minor corrections (see below).
Questions/Suggestions
Line 38: "Obstructive sleep apnea and central sleep apnea are the two primary forms of SDB observed in these patients, each with distinct pathophysiological mechanisms and implications for heart failure progression. "
Citation(s) needed
Line 43: "Reduced LVEF% in patients with OSA has been linked to increased left ventricular filling pressures and higher pulmonary artery pressures, which exacerbate heart failure symptoms and contribute to a worse prognosis. "
Citation(s) needed
Line 46: "Additionally, elevated E/e’ ratios in OSA patients suggest impaired diastolic function."
Line 48: "Patients with CSA often exhibit higher E/e’ ratios, indicating more pronounced diastolic dysfunction, and they tend to have elevated NT-proBNP levels due to the combined effects of ventricular wall stress and frequent episodes of hyperventilation and apnea. "
Citation(s) needed
Line 74: "in the cardiology department "
Please list hospital
Line 81: "During hospitalization and after signing informed consent"
Please list ethics approval information (committee, institution, ethics number) in methods section.
Line 82: "ApneaLinkTM somnographic screening system"
please provide more information on this, and/or citation. i.e Company information? how screen was conducted.
Line 116: Table 2.
Question. Were all P-values listed as 0.001 in table exactly 0.001, or P<0.001? If the latter please state this.
Please also include in table legend data is Mean ± SD
Line 123: "Patients were divided into two groups with and without home CPAP therapy after discharge."
Please include n-values of groups here
Line 127: "13 patients dropped out due to mortality."
Question: was cause of mortality determined?
Line 136: Figure 1; Line 142: Figure 2.; Line 155: Figure 3; Line 156: Figure 4.
Please include error bars (SD) on graphs
Please include in figure legends study length was 12 months
Please also include in figure legends data is Mean ± SD
Line 169: Figure 5.
Question: why change from bar graphs (figs 1-4) to individual plots in figure 5?
Author Response
Thank you very much for your review.
I have added all the necessary citations.
Question 1 - I have updated the text to reflect the statistical significance with 'p<0.001' as the correct notation.
Question: was cause of mortality determined?: Not in all patients, but the most common cause of mortality identified was cardiovascular mortality.
Question: why change from bar graphs (figs 1-4) to individual plots in figure 5?: We made this change because individual plots provide a clearer visualization of the data and enable us to monitor changes for each patient more precisely